# Key Factors of Quality Formation in Wuyi Black Tea during Processing Timing

**DOI:** 10.3390/foods13091373

**Published:** 2024-04-29

**Authors:** Li Lu, Jinxian Liu, Wenneng Zhang, Xi Cheng, Bo Zhang, Yiyang Yang, Youxiong Que, Yuanhua Li, Xinghui Li

**Affiliations:** 1College of Tea and Food, Wuyi University, Wuyishan 354300, China; 2Tea Engineering Research Center of Fujian Higher Education, Wuyishan 354300, China; 3Tea Science Research Institute, Wuyi University, Wuyishan 354300, China; 4Institute of Leisure Agriculture, Jiangsu Academy of Agricultural Sciences, Nanjing 210014, China; 5Key Laboratory of Sugarcane Biology and Genetic Breeding, Ministry of Agriculture and Rural Affairs, Fujian Agriculture and Forestry University, Fuzhou 350002, China; 6College of Horticulture, Nanjing Agricultural University, Nanjing 210095, China

**Keywords:** Wuyi black tea, processing, quality components, polyphenoloxidase, cellulase, gene expression

## Abstract

As the most consumed tea in the world, all kinds of black tea are developed from Wuyi black tea. In this study, quality components, regulatory gene expression, and key enzyme activity during the processing were analyzed to illustrate the taste formation of WBT. Withering mainly affected the content of amino acids, while catechins and tea pigments were most influenced by rolling and the pre-metaphase of fermentation. Notably, regulatory gene expression was significantly down-regulated after withering except for polyphenoloxidase1, polyphenoloxidase2, leucoanthocyanidin dioxygenase, chalcone isomerase, and flavonoid 3′, 5′-hydroxylase. Co-expression of flavonoid pathway genes confirmed similar expression patterns of these genes in the same metabolic pathway. Interestingly, rolling and fermentation anaphase had a great effect on polyphenol oxidase, and fermentation pre-metaphase had the greatest effect on cellulase. Since gene regulation mainly occurs before picking, the influence of chemical reaction was greater during processing. It was speculated that polyphenol oxidase and cellulase, which promoted the transformation of quality components, were the key factors in the quality formation of WBT. The above results provide theoretical basis for the processing of WBT and the reference for producing high-quality black tea.

## 1. Introduction

Black tea, which has health benefits, such as anti-oxidation, anti-radiation, anti-bacterial, and anti-inflammatory effects [1,2], is the most consumed tea in the world, accounting for 78% of the international tea trade [3]. Wuyi black tea (WBT) is represented by souchong black tea, which is originated in Wuyishan City, Fujian Province, China around 1567–1610 in the late Ming Dynasty. It is generally thought that the spread of WBT to the world has promoted the vigorous development of world black tea production, trade and consumption. In 1610, Xiaochong black tea was first exported to the Netherlands, and then exported to Britain, France, and Germany. After its introduction into the UK by the Dutch, the atmosphere of drinking black tea gradually spread in the UK from top to bottom and gradually evolved into an elegant afternoon tea culture, mostly due to promotion by the British royal family. It is not an exaggeration to say that the British spread the afternoon tea culture around the world, making black tea the largest tea category in the world.

Up to the present, abundant studies have been carried out to reveal biochemical components and their dynamic changes during the processing of black tea. Li et al. [4] investigated the dynamic changes of lipids during the processing and found significant changes in lipid patterns of black tea. The processing of black tea showed significant impact on biochemical compositions, i.e., significant decreases of catechins and tea polyphenols contents and significant increase of theaflavins levels [5,6]. Wu et al. [7] found that the biochemical components altered most significantly in the withering and rolling stages of black tea processing. Fermentation also plays an extremely important role. Research by Tan et al. [8] showed that chemical components such as catechins, amino acids, alkaloids, and other substances changed significantly through a series of chemical reactions during fermentation. However, as most studies focus on a certain kind of metabolite or a part of the processing, it is unclear how the quality of the components changes in throughout the processing of black tea, especially regarding different phases of withering and fermentation.

Wang et al. [9] found that the expression of seven raffinose synthetase genes was up-regulated at 12 h and down-regulated at 30 h during withering in white tea. Zhou et al. [10,11] demonstrated that the relative expression of the *CSA019598* gene was up-regulated after picking and reached the highest during shaking in oolong tea; particularly, two hypoxic stress response factors in gene expression were significantly up-regulated after hypoxic shaking tea treatment. From above, most research about gene expression mainly focused on white tea and oolong tea during processing. Only a few reports elaborated relevant gene expression in green tea and black tea. For instance, the expression of genes related to alkaloid synthesis was significantly up-regulated during the spreading of green tea [12]. Peroxidase (POD) and polyphenoloxidase (PPO) genes were expressed in different stages, and most ascorbate peroxidase (APX) genes were activated during black tea processing [13]. Furthermore, the expression of POD and PPO genes was up-regulated during withering in black tea under UV-C and ethylene treatment [14]. Regarding Wuyi black tea, the expression of genes related to quality component formation is still lacking.

PPO activity varies greatly in different stages during black tea processing [15], while the activities of PPO and cellulase (CL) increased during the pile fermentation in dark tea [16]. Notably, exogenous application of CL enhances theaflavin, thearubigin, and caffeine contents to be significantly higher than those of conventionally processed black tea [17]. However, there was no report on the changes of gene regulation and enzyme activity for the quality formation in WBT.

Generally, the taste of black tea is determined by flavonoids (including tea polyphenols, catechins, etc.) and their oxidation products (theaflavins, thearubigins, theabrownins, etc.), amino acids, alkaloids, and other chemical components, the changes in which have an important impact on the taste during processing [18,19]. Among them, flavonoids and their oxidation products are the main flavor substances of black tea. Additionally, the main chemical reaction is the enzymatic oxidation of tea polyphenols catalyzed by PPO. CL also plays an important role in promoting this reaction in this process. In the present study, we took samples at important nodes in the processing of WBT, and the main quality components of WBT were determined by national standard methods. Then, PPO and CL enzyme activities were detected with reagent kits. Furthermore, real-time fluorescence quantitative PCR (RT-qPCR) was used to measure gene expression. Combined with the correlation analysis, we explored the changes of relevant gene expression and enzyme activities during WBT processing and revealed their roles and relationships in the flavonoid metabolism pathway. This study was expected to provide theoretical basis and scientific perspective for high-quality black tea processing.

## 2. Materials and Methods

### 2.1. Materials and Chemicals

Thirty-year-old Fujian Shuixian tea trees (*Camellia sinensis* cv. Fujian-Shuixian), which were planted in the Science and Education Park of Wuyi University in Wuyishan City, Fujian Province, were used as the experimental material.

Reagents: methanol, ethanol, acetonitrile, acetic acid, n-butanol, ethyl acetate, disodium ethylenediamine tetraacetate, concentrated sulfuric acid, hydrochloric acid, ninhydrin, stannous chloride, ascorbic acid, dipotassium phosphate, potassium dihydrogen phosphate, aluminum trichloride (Sinopharm Chemical Reagent Co., Ltd., Shanghai, China). Standard substances: glutamic acid, caffeine (CAF), epigallocatechin (EGC), catechin (C), epicatechin (EC), epigallocatechin gallate (EGCG), gallocatechin (GC), gallic acid (GA), catechin gallate (CG), gallocatechin gallate (GCG), epicatechin gallate (ECG), theaflavin (TF), theaflavin-3-gallate (TF3G), theaflavin-3′-gallate (TF3′G), theaflavin-3, 3′-digallate (TFDG) (Xiamen Zhongpuda Instrument Co., Ltd., Xiamen, China). PPO and CL test kits (Solarbio Technology Co., Ltd., Beijing, China). Reverse transcription kit, SuperReal PreMix Plus and RNA extraction kit (Tiangen Biochemical Technology Co., Ltd., Beijing, China).

Instruments and equipment: UV-1880 ultraviolet visible spectrophotometer (Shanghai Mapada Instruments Co., Ltd., Shanghai, China); HH-4 digital display constant-temperature water-bathing boiler (Changzhou Guohua Electric Appliance Co., Ltd., Changzhou, China); Benchtop high-speed refrigerated centrifuge (Hunan Kaida technology Instrument Co., Ltd., Changsha, China); SHZ-DIII circulating water vacuum pump filter device (Zhengzhou Changcheng Science and Trade Co., Ltd., Zhengzhou, China); ZFD-A5040A air drying oven (Shanghai Jinghong Laboratory Instrument Co., Ltd., Shanghai, China); CFX96 real-time fluorescence quantitative PCR detection system (Bio-Rad Co., Ltd., Shanghai, China); Agilent 1260 high-performance liquid Chromatograph (Agilent Technologies Co. Ltd., Beijing, China).

### 2.2. Manufacturing Process and Tea Samples Collection

The fresh tea leaves were plucked with one bud and two leaves, and the processing technology of WBT was adopted. Technical flow process: withering → rolling → fermentation → drying.

Sampling Method 1 was used for determination of enzymatic activity (EA) and gene expression. Tea samples were collected at fresh leaves (FL), withering 6 h (WL-6), withering 12 h (WL-12), withering 18 h (WL-18), rolling (RL), fermenting 4 h (FL-4), and fermenting 8 h (FL-8) during processing, immediately frozen in liquid nitrogen, and stored at −80 °C.

Sampling Method 2 was used for determination of quality components. Seven tea samples collected at FL, WL-6, WL-12, WL-18, RL, FL-4, and FL-8 were dried twice with the dryer. The initial drying was 120 °C for 8 min, and the re-drying was 80 °C until fully dry. All samples were run in triplicate.

### 2.3. Determination of Quality Components and Enzyme Activities

The contents of water-soluble extracts and free amino acids were determined according to the national standards of GB/T 8305-2013 [20] and GB/T 8314-2013 [21], respectively. The contents of tea polyphenols (TPs), catechins, CAF, TF, TF3G, TF3′G, and TFDG were determined according to the national standard GB/T8313-2018 [22]. The contents of theaflavins (TFs), thearubigins (TRs), and theabrownins (TBs) were determined using the spectra photometric method described by Wang [23]. The total amount of flavonoids was determined using the colorimetric method. The 0.5 mL of tea extract solution and 10 mL of 1% aluminum chloride (*w*/*v*) were added into a 100 mL triangular flask. The solution was mixed well, and after 10 min, the absorbance was measured immediately against the prepared blank at 420 nm using a UV-1880 ultraviolet–visible spectrophotometer. The determination of PPO and CL activities referred to the test kits instructions, respectively. All samples were analyzed in triplicate.

### 2.4. Determination of Gene Expression

RNA extraction and cDNA synthesis were conducted according to the SuperReal PreMix Plus and RNA extraction kit and reverse transcription kit instructions, respectively. Glyceraldehyde-3-phosphate dehydrogenase (GAPDH) was used as the internal reference gene, and gene expressions of APX, PPO1, PPO2, phenylalanine ammonialyase (PAL), cinnamate 4-hydroxylase (C4H), chalcone synthase (CHS), chalcone isomerase (CHI), flavanone 3-hydroxylase (F3H), flavonoid-3′-hydroxylase (F3′H), flavonoid 3′, 5′-hydroxylase (F3′5′H), flavonol synthase (FLS), dihydroflavonol 4-reductase (DFR), leucoanthocyanidin dioxygenase (LDOX), leucoanthocyanidin reductase (LAR), and anthocyanidin reductase (ANR) were determined. Primer design is shown in Table 1. Reaction system: 20 μL; first-strand cDNA template 1.0 μL, Primer-F 0.5 μL, Primer-R 0.5 μL, 2×NovoStart SYBR qPCR SuperMix Plus 10.0 μL, dd H_2_O 8.0 μL. Reaction procedure: denatured at 95 °C for 3 min; denatured at 95 °C for 20 s; annealing at 55 °C for 20 s; 72 °C extension 30 s, 40 cycles; extend for 5 min at 72 °C. qPCR data processing method: The relative expression quantity was used in this experiment, and 2-ΔΔCt was selected as the calculation method. All samples were tested in triplicate.

### 2.5. Statistical Analyses

The proportion of a certain quality component in tea sample was calculated as follows:$$\mathrm{Proportion}(\%)=\frac{A-B}{C}\times 100$$

In the formula, *A* represents the content of a certain quality component at a certain processing node; *B* represents the content of a certain quality component at the previous processing node; *C* represents the total absolute value of changes in the content of a certain quality component at each processing node; the proportion refers to the proportion of change in a certain processing node to the total processing of a certain quality component. Positive proportion indicates increasing effect, and negative proportion indicates decreasing effect.

All the results were carried out in triplicate for the analytical determination. To assess differences among different processing nodes, DPS 7.05 was used to perform the least significant difference (LSD), Duncan’s multiple ranges, and Student’s t tests. Statistics were deemed significant at *p* value < 0.05. All data are presented as the mean ± standard deviation (SD). Pearson correlation analysis was performed by SPSS 18.0. Figures were drawn using Origin 8.5, HemI 1.0 and Adobe Photoshop CS6 13.0 statistical software.

## 3. Results

### 3.1. Dynamic Changes in Quality Components during WBT Processing

As shown in Figure 1A,B, the content of water-soluble extracts gradually decreased but kept relatively stable in withering, and there were significant differences between various stages during WBT processing. The content of free amino acids increased first and then decreased and reached a highest value of 4.89% (*w*/*w*) in WL-18. The content of caffeine did not change obviously and reached a highest value of 3.48% (*w*/*w*) in WL-18. The flavonoids content showed an increasing trend on the whole, reaching a highest value of 1.16% at FL-8. Except for WL-12, there were significant differences between other nodes, indicating that the flavonoids content changed greatly during WBT processing. The content of TPs showed a decreasing trend to the lowest 15.90% at FL-8. Eight catechin monomers, ester catechins, simple catechins and total catechins all declined rapidly after rolling (Figure 1C–E). The variation of GA was opposite to that of eight catechin monomers and showed a significant increase after rolling, which may be attributed to the hydrolysis of EGCG for the large formation of GA. The contents of ester catechins, simple catechins, and total catechins were all the lowest in FL-4 at 14.32 mg/g, 4.71 mg/g, and 19.02 mg/g, respectively. As shown in Figure 1F,G, tea pigments showed an overall upward trend and began to rise sharply after rolling. TFDG and TF3′G increased to the highest at FL-4. Theaflavins increased linearly from WL-18 to FL-4 and then tended to flatten out. The growth rate of thearubigins slowed down after FL-4, while theabrownines still increased rapidly. The contents of thearubigins and theabrownines were the highest in FL-8, at 6.47% (*w*/*w*) and 7.65% (*w*/*w*), respectively.

### 3.2. Great Effect of Processing Nodes on the Formation of Quality Components

As shown in Figure 2, WL-12 and WL-18 had the greatest effect on amino acids, accounting for 16.39% and 24.47%, respectively. RL had the greatest effect on water-soluble extracts, accounting for 40.98%, followed by amino acids and flavonoids, accounting for 27.02% and 24.22%, respectively. The effects of FL-4 on flavonoids, caffeine, tea polyphenols and water-soluble extracts were the highest in all processing nodes, accounting for 49.17%, 48.47%, 40.90%, and 33.03%, respectively. FL-8 had a great impact on tea polyphenols and flavonoids, accounting for 19.18% and 13.40%, respectively (Figure 2A). However, WL-6, WL-12, and WL-18 had little effect on catechins. The contents of all catechins decreased by FL-4. RL and FL-4 had the greatest effect on catechins, together accounting for more than 80%. Among them, GA, GC and EGC accounted for 61.84%, 53.9%, and 53.8% by RL, respectively. The proportions of C, EC, GCG, CG, ECG and ester catechins impacted by FL-4 were 50.81–64.59%. FL-8 had a great effect on GA, C, CG and ECG, accounting for more than 10% (Figure 2B). WL-6, WL-12, and WL-18 had little effect on tea pigments. RL increased the content of all tea pigments. RL and FL-4 had the greatest effect on tea pigments, which accounted for more than 72% when combined. Among them, TF, TF3′G and total theaflavins impacted by RL accounted for 77.52%, 68.24%, and 48.31%, respectively. The proportions of TFDG, TF3G, total theaflavins, thearubigins, and theabrownines impacted by FL-4 were between 45.41% and 57.96% (Figure 2C).

### 3.3. Dynamic Changes in the Expression of Genes in Flavonoid Metabolic Pathway during WBT Processing

RT-qPCR analysis showed the relative expression levels of 15 genes in flavonoid metabolism pathway (12 genes, such as PAL, C4H, CHS, CHI, F3′5′H, F3H, DFR, LAR, LDOX, ANS, ANR, etc.), APX, PPO1 and PPO2 (Figure 3). PAL, C4H, CHS, F3′H, F3H, DFR, LAR, ANS, ANR, and APX were significantly down-regulated during processing (Figure 3). LDOX was also down-regulated during processing, but the expression trend was different from that of the other 14 genes. Surprisingly, the expression trends of PAL, C4H, CHS and DFR genes were consistent. Specifically, the expressions of CHI and F3′5′H genes showed the same trend, being up-regulated in WL-6 1.05 times and 1.34 times more than fresh leaves, respectively, and significantly down-regulated in subsequent nodes. In addition, the expression of PPO1 and PPO2 was up-regulated → down-regulated → up-regulated → down-regulated → up-regulated, and their expression levels in WL-18 were 6.37 times and 5.59 times of those in fresh leaves, respectively, while the expression levels in FL-8 were 17.74 times and 3.43 times of those in fresh leaves, respectively.

### 3.4. Dynamic Changes in Key Enzyme Activities during WBT Processing

PPO is the key enzyme in black tea fermentation, and closely related to the formation of theaflavins, thearubigins, and theabrownines [15]. As shown in Figure 4, PPO activity first increased and then decreased, followed by an increase and then a decrease with two peaks at WL-12 and RL, but rapidly declined during fermentation. The overall trend of CL activity first increased and then decreased. The activity of CL increased gradually from FL to RL, which reached the highest at the end of rolling and then decreased rapidly after fermentation. PPO and CL activities were increased by WL-6, WL-12, and RL. CL activity was increased, and PPO activity was decreased by WL-18. The activities of PPO and CL were decreased by FL-4 and FL-8. RL and FL-8 had great effect on PPO, accounting for 23.26% and 23.72%, respectively, followed by FL-4. FL-4 had the greatest effect on CL, accounting for 37.41%, followed by FL-8 and RL (Figure 2D).

### 3.5. Close Relationship among Gene Expression Levels during WBT Processing

In view of the low relative expression levels of 4 genes—F3H, F3′H, LAR, and APX—during WBT processing, we only conducted correlation analysis for the other 11 genes. PPO1, PPO2 and LDOX had no significant correlation with all the detected genes (Figure 5). F3′5′H was only significantly correlated with CHI at the 0.01 level, and CHI was only significantly correlated with PAL and F3′5′H at the 0.05 level. PAL, CHS, C4H, FLS, DFR, and ANR were significantly correlated with each other at the 0.01 level. The fact that most genes of the flavonoid pathway were closely related to each other, indicating that gene groups of the same metabolic pathway tend to be similar in expression. Almost no significant correlation was found between enzyme activity and gene expression when analyzed with different nodes simultaneously, and the same phenomenon existed when correlation was analyzed between quality components and enzyme activity or gene expression. As shown in Figure 5, PPO activity was significantly negatively correlated with PPO1 and CHS at the 0.01 level but not with other genes. Additionally, the activity of CL was negatively correlated with PPO1 and LDOX at the 0.05 level but not with other genes.

### 3.6. Close Relationship among the Quality Components, Enzyme Activity, and Gene Expression during WBT Processing

It can be seen from Figure 6 that the correlation between PPO or CL activities and quality components is basically consistent—that is, they were all significantly correlated with most quality components. Specifically, PPO and CL activities were significantly and positively correlated with the content of water-soluble extracts, caffeine, GC, EGC, EGCG, EC, ECG, ester catechins, simple catechins, and total catechins at the 0.05 level and positively correlated with GCG at the 0.01 level. PPO activity was significantly and negatively correlated with the contents of flavonoids, TFDG, TF3G, total theaflavins, thearubigins, and theabrownins at the 0.05 level. Meanwhile, there was a significant negative correlation between CL activity and catechin oxidation products at the 0.05 level. We thus speculated that the quality components might have a reverse effect on enzyme activity, and catechin oxidation products inhibited PPO and CL activities, which was consistent with the changes of enzyme activities shown in Figure 4. However, contrary to the changing rule of enzyme activity, PPO1 was significantly positively correlated with TFDG, TF3G, total theaflavins, thearubigins, and theabrownins at the 0.05 level; significantly and positively correlated with TF3′G and flavonoids at the 0.01 level; and significantly and negatively correlated with water soluble extracts, tea polyphenols, and catechins. Furthermore, ANR was positively correlated with tea polyphenols at the 0.05 level, and negatively correlated with GA and TF at the 0.05 level. We also found that other genes were not significantly associated with each quality component. In addition, PPO1 was significantly correlated with polyphenol oxidation product, PPO, and CL activities, suggesting that PPO1 was one of the key genes in the quality formation of WBT. However, both PPO and CL activities were inhibited by the oxidation products while promoting the enzymatic oxidation reaction.

## 4. Discussion

During the processing of black tea, the contents of tea polyphenols and catechins showed decreasing trends in general, while the total catechins content decreased significantly in the fermentation stage. Additionally, the content of GA increased, and theaflavins showed a trend of first increasing and then decreasing [13,24,25], which is similar with the results of our study. Previous research has revealed that catechins, most abundant in fresh leaves, reduced significantly during processing, especially in the late rolling and fermentation stages [6,24,26]. Deka et al. [5] reported that tea polyphenol content reduced rapidly and that the content of EGCG reduced more than that of ECG during fermentation. Owuor and Obanda [27] found that the oxidation rate of trihydroxycatechins (EGCG and EGC) was faster than that of their dihydroxyl counterparts (ECG and EC). Meanwhile, theaflavins was produced first, followed by thearubigins and theabrownins [28]. Teng et al. [29] confirmed that high concentrations of EGCG and ECG led to an increase in TFDG synthesis in a model study using PPO enzyme reaction to synthesize theaflavins. The cleavage of gallic groups may be one of the factors contributing to the decrease of ECG content before theaflavin formation [2]. Theaflavin content decreased after reaching the maximum, which can be attributed to the conversion of theaflavins to thearubigins and catechins with low level [30,31,32]. These changes of catechins and tea pigments were consistent with the results of our study.

In the present study, due to protein hydrolysis in withering, amino acid content increased and then decreased to convert into other substances, such as aromatic substances in the late processing of black tea, which was consistent with the results of Qu et al. [33] and Chen et al. [34]. Caffeine and flavonoids are considered as the main bitterness substances in tea. There are several ways to increase caffeine: firstly, caffeine synthesis in the withering stage; secondly, there may be a relation to amino acid metabolism; thirdly, the decomposition of nucleic acid leads to the production of caffeine [5,35,36]. For the pathway of caffeine reduction, Wang et al. [37] found that purine alkaloids may degrade into imidazole. Sari and Velioglu [38] reported that caffeine content increased after withering and decreased after drying. Tan et al. [8] discovered that the caffeine content remained relatively stable during fermentation, while Kim et al. [2] observed that caffeine content reduced during fermentation. It should be pointed out here that tea varieties and processing technology may lead to the discordance of caffeine change among previous studies. Further studies have found that caffeine increases significantly in withering and remains relatively stable in subsequent stages [39,40,41,42], which was consistent with our study.

We can reasonably deduce that the products from gene regulation lay the material basis of tea fresh leaves. Except for five genes, such as F3′5′H, most of the other genes were significantly down-regulated from withering in our study. Zhang et al. [13] found that most genes were significantly down-regulated during the processing of black tea except that the expression of F3′5′H was up-regulated and C4H was down-regulated first and then up-regulated in the fermentation stage, which was basically consistent with the results in our study. However, different from the report of Zhang et al. [13], in which the APX gene was still highly expressed during the processing of black tea, the APX gene was almost not expressed except in FL due to the difference in tea varieties and processing technology. In particular, similar to the results of Luo et al. [43], almost no significant correlation was found between quality components and gene expression. The gene regulation mainly occurs before picking. However, the expression of most genes was rapidly down-regulated after fresh leaf harvesting in vitro. The intermediate process from gene expression to product generation takes time, so genes related to flavonoid pathway did not directly affect product accumulation. Notably, chemical reactions during processing caused dramatic changes of the content of some quality components, and the impact of chemical reactions was greater than that of gene regulation. Unexpectedly, several quality components were significantly correlated with gene expression and enzyme activity, and whether or not this implies that gene and protein expression synergistically regulate the production of quality components in black tea still needs further study.

Interestingly, exploring the changes in key endogenous enzymes that affect the flavor components of black tea is helpful to improve its flavor. PPO is a metalloenzyme with type III copper as its core, which is abundant in nature [44]. Compared to POD and glucosidase, PPO, which can catalyze the oxidation of catechins to theaflavins and thearubigins as well as the enzyme coupled oxidation of aromatic precursors [45], plays an important role in the catalysis of flavonoids during tea processing [46]. Previous studies have shown that PPO activity reached the highest in rolling, up to 2–3 times that of fresh leaves, and decreased during fermentation and drying [47,48]. Our study demonstrated that PPO activity first increased and then decreased during withering, reached its peak in rolling, and then continuously decreased, which was basically consistent with the results of previous studies [45,47,48]. The possible reason was that as the withering time progressed, the water in leaves gradually dispersed, the concentration of cell fluid increased, and the activities of some hydrolase, and oxidase increased. Then, the withering leaves gradually became acidic due to the degradation of carbohydrates into organic acid, the degradation of ester catechins into gallic acid, the degradation of protopectin into pectinic acid, and the hydrolysis of chlorophyll into chlorophyllin acid. Thus, the pH of the fresh leaves decreased from nearly neutral to the optimal pH of PPO and the enzyme activity increased. Additionally, the organic acids accumulated continue to reduce pH late in the withering stage, which prevented the enzyme from performing its function optimally, while water stress and protein degradation also contributed to the decrease of PPO activity in leaves. This may be the reason for the decrease of enzyme activity in late withering [44,49]. Jiang et al. [28] showed that the increase and then decrease in PPO activity was not completely consistent with the change trend found in this study, which might be related to variety, season and appellation. CL is a multi-component complex system of enzymes [50]. Previous studies showed that the addition of exogenous CL could help to destroy the cell wall in tea; promote the release of the substances such as tea polyphenols, amino acids, and caffeine contained in rolling leaves; facilitate the full transformation of tea polyphenols; enhance the water soluble extracts and theaflavin content; and improve the quality and aroma of black tea [13]. In this study, CL activity increased gradually during withering, and the principle might be similar to that of PPO—that is, the content of enzyme substrate decreased in fermentation, resulting in a continuous decline in enzyme activity.

In summary, we depicted here a potential model for quality formation, regulatory gene expression, and key enzyme activities in Wuyi black tea during processing (Figure 7).

## 5. Conclusions

From all the above, the majority of quality components changed dramatically during WBT processing and mainly occurred in the two stages of RL and FL-4. It is thus clear that the key period for the formation of WBT quality was rolling and fermentation pre-metaphase. The mechanism of dynamic change of some quality components, such as caffeine, has been partially confirmed, but it was not completely clear and needs to be further studied. The activities of PPO and CL changed most during rolling and fermentation. However, there was no significant correlation between quality components and enzyme activity through synchronous correlation analysis, which thus speculated that more complex factors might be involved except enzymatic catalysis. Nevertheless, the enzyme activity reached the highest in rolling, while the main changes of tea polyphenols and their oxidized products were completed in rolling and subsequent fermentation pre-metaphase, indicating that enzyme activity promoted the reaction as a key factor for the quality formation of WBT. These above results could provide theoretical basis for exploring the processing principle of WBT and set up the practical reference for producing high-quality black tea.

## Figures and Tables

**Figure 1 foods-13-01373-f001:**
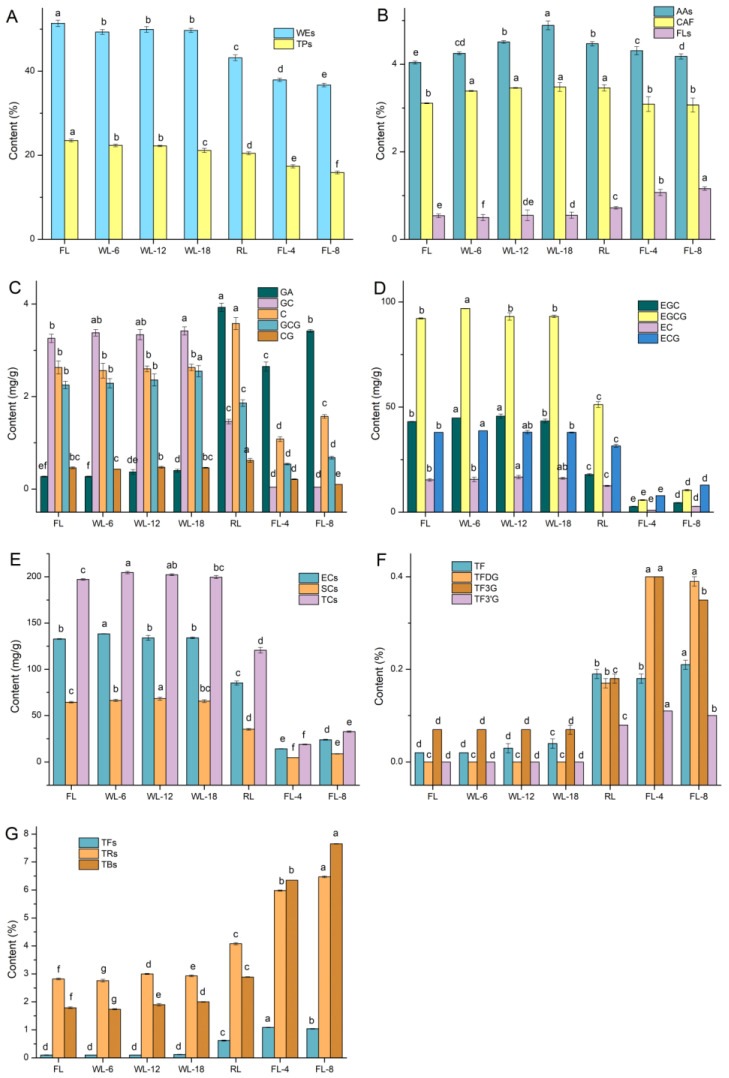
Contents of quality components during WBT processing. (**A**,**B**), quality compounds; (**C**–**E**), catechins; (**F**,**G**), tea pigments. Data are presented as the mean ± SD (standard deviation) and were assessed by one-way ANOVA. The different lowercase letters indicate the least significant difference (*p* < 0.05). WEs, water-soluble extracts; TPs, tea polyphenols; AAs, amino acids; CAF, caffeine; FLs, flavonoids; GA, gallic acid; GC, gallocatechin; C, catechin; GCG, gallocatechin gallate; CG, catechin gallate; EGC, epigallocatechin; EGCG, epigallocatechin gallate; EC, epicatechin; ECG, epicatechin gallate; ECs, ester catechins; SCs, simple catechins; TCs, total catechins; TF, theaflavin; TFDG, theaflavin-3, 3′-digallate; TF3G, theaflavin-3-gallate; TF3′G, theaflavin-3′-gallate; TFs, theaflavins; TRs, thearubigins; TBs, theabrownines; FL, fresh leaves; WL-6, withering 6 h; WL-12, withering 12 h; WL-18, withering 18 h; RL, rolling; FL-4, fermenting 4 h; FL-8, fermenting 8 h.

**Figure 2 foods-13-01373-f002:**
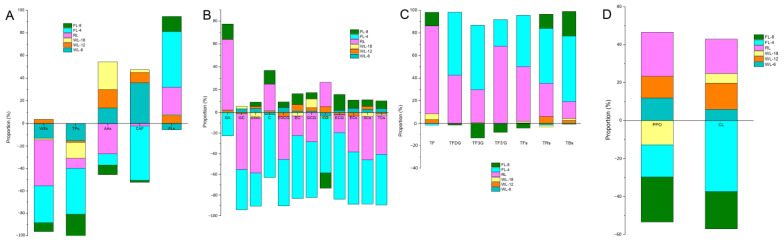
Proportion of processing nodes to the formation of quality. (**A**), quality compounds; (**B**), catechins; (**C**), tea pigments; (**D**), enzyme activities. WEs, water-soluble extracts; TPs, tea polyphenols; AAs, amino acids; CAF, caffeine; FLs, flavonoids; GA, gallic acid; GC, gallocatechin; C, catechin; GCG, gallocatechin gallate; CG, catechin gallate; EGC, epigallocatechin; EGCG, epigallocatechin gallate; EC, epicatechin; ECG, epicatechin gallate; ECs, ester catechins; SCs, simple catechins; TCs, total catechins; TF, theaflavin; TFDG, theaflavin-3, 3′-digallate; TF3G, theaflavin-3-gallate; TF3′G, theaflavin-3′-gallate; TFs, theaflavins; TRs, thearubigins; TBs, theabrownines; FL, fresh leaves; WL-6, withering 6 h; WL-12, withering 12 h; WL-18, withering 18 h; RL, rolling; FL-4, fermenting 4 h; FL-8, fermenting 8 h.

**Figure 3 foods-13-01373-f003:**
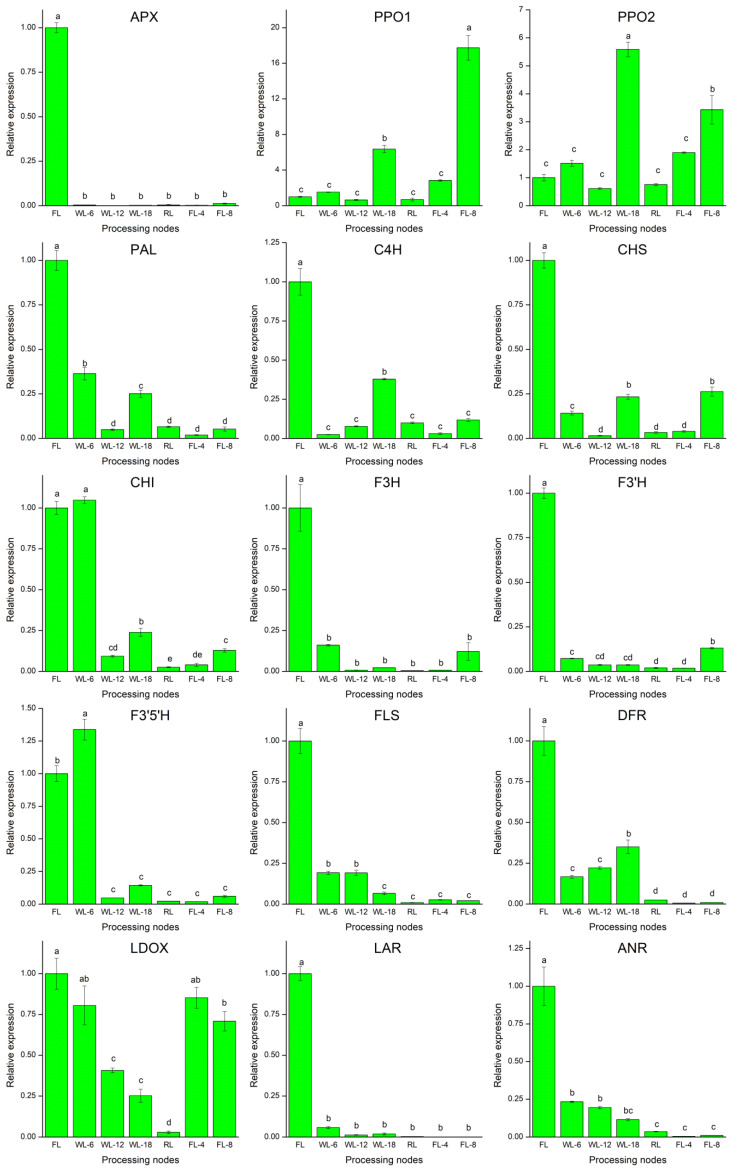
Relative expression of regulatory gene during the processing of WBT. Data are presented as the mean ± SD (standard deviation) and were assessed by one-way ANOVA. The different lowercase letters indicate the least significant difference (*p* < 0.05). APX, ascorbate peroxidase; PPO1, polyphenol oxidase 1; PPO2, polyphenol oxidase 2; PAL, phenylalanine ammonialyase; C4H, cinnamate 4-hydroxylase; CHS, chalcone synthase; CHI, chalcone isomerase; F3H, flavanone 3-hydroxylase; F3′H, flavonoid-3′-hydroxylase; F3′5′H, flavonoid3′, 5′-hydroxylase; FLS, flavonol synthase; DFR, dihydroflavonol 4-reductase; LDOX, leucoanthocyanidin dioxygenase; LAR, leucoanthocyanidin reductase; ANR, anthocyanidin reductase; FL, fresh leaves; WL-6, withering 6 h; WL-12, withering 12 h; WL-18, withering 18 h; RL, rolling; FL-4, fermenting 4 h; FL-8, fermenting 8 h.

**Figure 4 foods-13-01373-f004:**
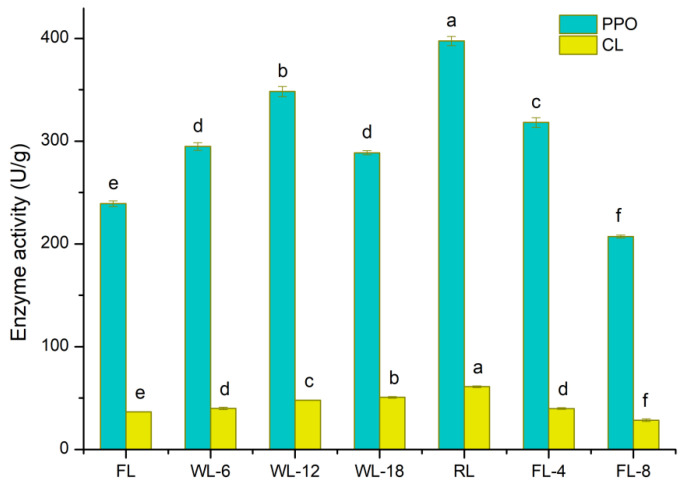
Activities of PPO and CL during the processing of WBT. Data are presented as the mean ± SD (standard deviation) and were assessed by one-way ANOVA. The different lowercase letters indicate the least significant difference (*p* < 0.05). PPO, polyphenol oxidase; CL, cellulase; FL, fresh leaves; WL-6, withering 6 h; WL-12, withering 12 h; WL-18, withering 18 h; RL, rolling; FL-4, fermenting 4 h; FL-8, fermenting 8 h.

**Figure 5 foods-13-01373-f005:**
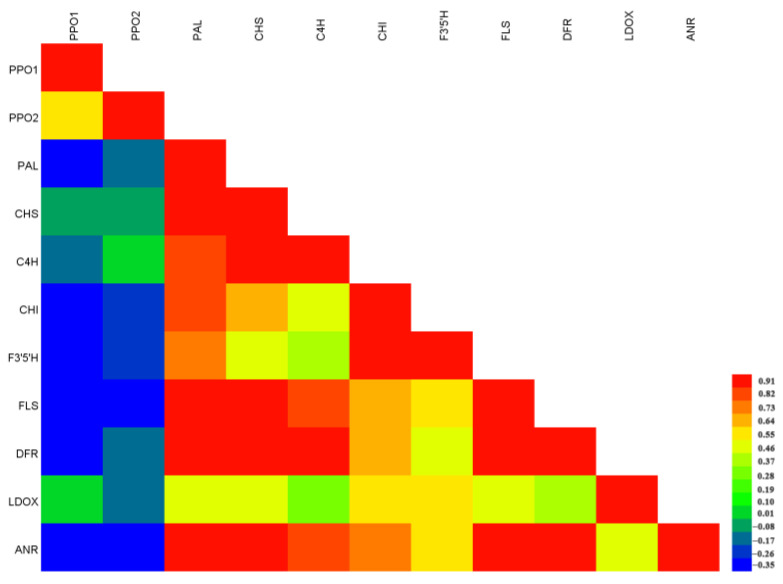
Correlation analysis among gene expression levels during WBT processing.

**Figure 6 foods-13-01373-f006:**
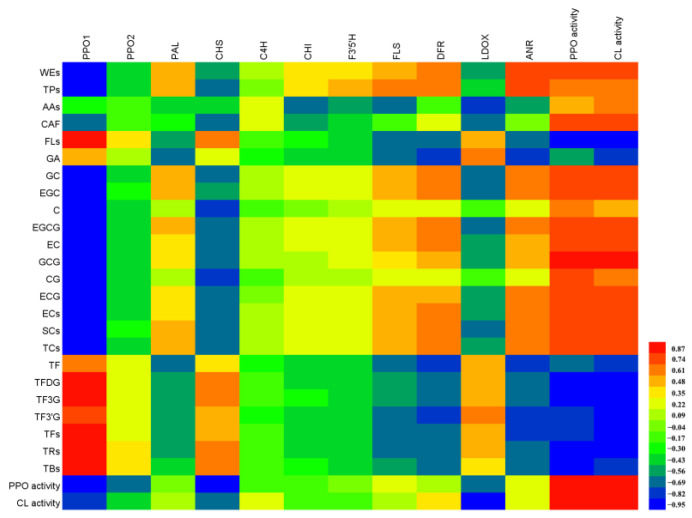
Correlation analysis between quality components, regulatory gene expression, and enzyme activities during WBT processing.

**Figure 7 foods-13-01373-f007:**
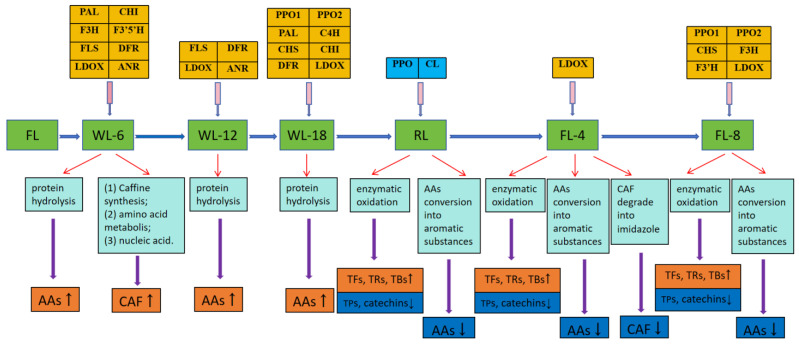
A potential model for quality formation, regulatory gene expression and key enzyme activities in Wuyi black tea during processing.

**Table 1 foods-13-01373-t001:** Primers for qPCR amplification.

Gene	Primer Sequence (5′→3′)	Primer Sequence (3′→5′)
GAPDH	F:TTGGCATCGTTGAGGGTCT	R:CAGTGGGAACACGGAAAGC
APX	F:TTCTATCAGTTGGCTGGAGTTG	R:AATGGTCACATCCCTTATCGG
PPO1	F:CCATCTGGAAGAGTTTGGGT	R:CCTTCACTTTGACAGGCTGA
PPO2	F:ATCTGGAAGAGTTTGGGTAGGC	R:CACTTTGACAGGCTGAGCATTC
PAL	F:CAATAGGGAAGCTCATGTTTGC	R:CGCTTTGGACATGGTTGGTTAC
C4H	F:TCACCGAGCCAGACACCTAC	R:CCTTAGCCTCCTCTTCCAAGA
CHS	F:GTGGGCCTTACATTTCATCTC	R:TCTAGTATGAATAGCACGCAC
CHI	F:GTTAAGTGGAAGGGCAAGAC	R:GAAAGCAATCGTCAATGATCC
F3H	F:TCTACCCGAAATGCCCACAAC	R:CCTCCCATTGCTTAGATAATG
F3′H	F:TCGAATGGCATCTGACAGTTG	R:GCCTGCACCAAATGGTATGAC
F3′5′H	F:GAGCACACGACGAGATGGAT	R:GTCTTTGCATTCTTTCCACTC
FLS	F:GCATGAGGTCAAGGAGGCTGT	R:GACAATCAGGGCATTAGGGATG
DFR	F:CACTAGGAATGAAGGACACTAC	R:GAACGACACAACTGGCAAGT
LDOX	F:CAGTAATCCGTGTTCAATCCTTG	R:TAAACCTGCTTCTCTTCCATG
LAR	F:CTATGACAATACTCACCCATC	R:GAGTGCGTCCAATCTTCTTCT
ANR	F:TCGAAAACACTAGCTGAGAAAG	R:GCTCGGGAACACTGGTATTG

## Data Availability

The original contributions presented in the study are included in the article, further inquiries can be directed to the corresponding author.

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
