# Peer review of "Key Factors of Quality Formation in Wuyi Black Tea during Processing Timing"

_foods, 2024, doi:10.3390/foods13091373_

Round 1
Reviewer 1 Report
Comments and Suggestions for Authors
A lot of work has been done by the authors. I believe that the topic is interesting and that the experimental procedure is valid. However, the manuscript has several important shortcomings.
The abstract should be reorganized. A short background should be provided. It is not clear which activities have been performed and which are the main findings.
The introduction is a simple list of previous studies. I believe that it should be reorganized and made more thorough in their argumentation.
The material and methods section contains unuseful information (e.g. the companies of the reagents). Some paragraphs are confusing and some procedures are not clear. What was the age of the sampled plants? How many biological replicates were collected? “RNA extraction and cDNA synthesis were conducted refer to the kit instructions.” Which kit?
The Results should be described in more detail. The high number of abbreviations makes the results difficult to understand. The figure captions should describe in detail the content and all the abbreviations of the figure.
The conclusions should focus on the main findings and their impact in a concise way. I believe that is too long and repetitive of the results.
Overall, the manuscript contains valuable information, but, in my opinion, it needs to be significantly reorganized and improved.
Comments on the Quality of English LanguageI strongly suggest to have the work reviewed by a native English speaker.
Author Response
Response to Reviewer 1 Comments
Point-by-point response to Comments and Suggestions for Authors
Comments 1: The abstract should be reorganized. A short background should be provided. It is not clear which activities have been performed and which are the main findings.
Response 1: Thank you for pointing this out. We agree with this comment. Therefore, I have reorganized the abstract. A short background was added in line 14. Some non-major findings were edited out, and the sentence in line 15-17 was modified.
Comments 2: The introduction is a simple list of previous studies. I believe that it should be reorganized and made more thorough in their argumentation.
Response 2: Agree. In introduction, we have reorganized the previous studies, and made more thorough in their argumentation. We divide the three aspects including biochemical components, gene expression and enzymatic activity during the processing of black tea into three paragraphs to elaborate in paragraphs 2-4.
Comments 3: The material and methods section contains unuseful information (e.g. the companies of the reagents). Some paragraphs are confusing and some procedures are not clear. What was the age of the sampled plants? How many biological replicates were collected? “RNA extraction and cDNA synthesis were conducted refer to the kit instructions.” Which kit?
Response 3: Thank you for pointing this out. We have deleted the companies of the reagents, and re-segmented some paragraphs. The information of age of the sampled plants, biological replicates and the definite kit instructions was added in line 101, 131, 141-142, 144-145 and 158, respectively.
Comments 4: The Results should be described in more detail. The high number of abbreviations makes the results difficult to understand. The figure captions should describe in detail the content and all the abbreviations of the figure.
Response 4: Thank you for pointing this out. We have described the results in more detail. We have reduced the use of abbreviations. The information of lowercase letters, standard deviation, significance level and abbreviations was added in figure captions.
Comments 5: The conclusions should focus on the main findings and their impact in a concise way. I believe that is too long and repetitive of the results.
Response 5: We agree with this comment. We reorganized some of the conclusions in line 432-433 and delete some duplicate results to make the conclusions more concise.
2
3. Response to Comments on the Quality of English Language
Point 1: I strongly suggest to have the work reviewed by a native English speaker.
Response 1: Thanks very much for your comments, which are very helpful for us to improve the manuscript. After carefully check, we found many grammar and sentence errors, and have modified the manuscript accordingly. Furthermore, we have invited several English teachers to help correct grammar and sentences with the hope that the revised paper will be more clear and accurate on expressions.
Reviewer 2 Report
Comments and Suggestions for Authors
Manuscript "Key factors of quality formation in Wuyi black tea during processing timing" is well written and provides new knowlegde about changes of phytochemical composition as well as gene expression during processing of tea. Obtained results could be used in practice. Authors provides some recommendations for the future researches. Howerver, I have some comments for the authors:
a) 2.5 Statistical analyses: there should be provided information about replications and what error bars in the diagrams show. Also, there should be information which tests were used to evaluate differences between samples and which test was used to evaluate correlation as well as significance level.
b) Line 138-140: I couldn't fined mentioned national standards in cited reference 19.
c) Line 140-141: colorimetric method should be described in detail. There is no exact information about method in reference 21.
d) Figure 1: there should be more provided information: means of abbreviations, letters above the columns, also what is depicted in A-G figures.
e) Figure 2: the means of abbreviations should be included.
f) Figure 3: the means of abbreviations and letters above the columns should be included.
g) Names of results subsections should be corrected. Now they are statements.
h) Figure 7: maybe it should be included to the discussion section, not to the conclusions.
Author Response
|
Response to Reviewer 2 Comments |
||
|
Point-by-point response to Comments and Suggestions for Authors |
||
|
Comments 1: 2.5 Statistical analyses: there should be provided information about replications and what error bars in the diagrams show. Also, there should be information which tests were used to evaluate differences between samples and which test was used to evaluate correlation as well as significance level. |
||
|
Response 1: Thank you for pointing this out. We agree with this comment. Therefore, we have added information about replications, error bars, the test used and significance level in line 169-173. |
||
|
Comments 2: Line 138-140: I couldn't fined mentioned national standards in cited reference 19. |
||
|
Response 2: There is clear information in national standards, so we have deleted reference 19. |
||
|
Comments 3: Line 140-141: colorimetric method should be described in detail. There is no exact information about method in reference 21. |
||
|
Response 3: Thank you for pointing this out. The exact information of colorimetric method has been added in detail in line 137-140, and we have deleted reference 19. |
||
|
Comments 4: Figure 1: there should be more provided information: means of abbreviations, letters above the columns, also what is depicted in A-G figures. |
||
|
Response 4: Thank you for pointing this out. The information of means of abbreviations, letters above the columns, and description of A-G figures was added in Figure 1. |
||
|
Comments 5: Figure 2: the means of abbreviations should be included. |
||
|
Response 5: The means of abbreviations have been added in Figure 2. |
||
|
Comments 6: Figure 3: the means of abbreviations and letters above the columns should be included. |
||
|
Response 6: The means of abbreviations and letters above the columns have been added in Figure 3. |
||
|
Comments 7: Names of results subsections should be corrected. Now they are statements. |
||
|
Response 7: We agree with this comment. We have modified the names of results subsections. |
||
|
Comments 8: Figure 7: maybe it should be included to the discussion section, not to the conclusions. |
||
|
Response 8: We agree with this comment. We've moved Figure 7 from the conclusion to the discussion. |